# Scoliosis Management through Apps and Software Tools [note 1]

**DOI:** 10.3390/ijerph20085520

**Published:** 2023-04-14

**Authors:** Lorella Bottino, Marzia Settino, Luigi Promenzio, Mario Cannataro

**Affiliations:** 1Department of Medical and Surgical Sciences, University Magna Graecia of Catanzaro, 88100 Catanzaro, Italy; 2Pediatric Orthopaedics Department, Villa Serena for Children, 88100 Catanzaro, Italy

**Keywords:** scoliosis, scoliosis management, scoliometer, Cobb angle, angle of trunk rotation (ATR), axial vertebral rotation (AVR), app

## Abstract

**Background:** Scoliosis is curvature of the spine, often found in adolescents, which can impact on quality of life. Generally, scoliosis is diagnosed by measuring the Cobb angle, which represents the gold standard for scoliosis grade quantification. Commonly, scoliosis evaluation is conducted in person by medical professionals using traditional methods (i.e., involving a scoliometer and/or X-ray radiographs). In recent years, as has happened in various medicine disciplines, it is possible also in orthopedics to observe the spread of Information and Communications Technology (ICT) solutions (i.e., software-based approaches). As an example, smartphone applications (apps) and web-based applications may help the doctors in screening and monitoring scoliosis, thereby reducing the number of in-person visits. **Objectives:** This paper aims to provide an overview of the main features of the most popular scoliosis ICT tools, i.e., apps and web-based applications for scoliosis diagnosis, screening, and monitoring. Several apps are assessed and compared with the aim of providing a valid starting point for doctors and patients in their choice of software-based tools. Benefits for the patients may be: reducing the number of visits to the doctor, self-monitoring of scoliosis. Benefits for the doctors may be: monitoring the scoliosis progression over time, managing several patients in a remote way, mining the data of several patients for evaluating different therapeutic or exercise prescriptions. **Materials and Methods:** We first propose a methodology for the evaluation of scoliosis apps in which five macro-categories are considered: (i) technological aspects (e.g., available sensors, how angles are measured); (ii) the type of measurements (e.g., Cobb angle, angle of trunk rotation, axial vertebral rotation); (iii) availability (e.g., app store and eventual fee to pay); (iv) the functions offered to the user (e.g., posture monitoring, exercise prescription); (v) overall evaluation (e.g., pros and cons, usability). Then, six apps and one web-based application are described and evaluated using this methodology. **Results:** The results for assessment of scoliosis apps are shown in a tabular format for ease of understanding and intuitive comparison, which can help the doctors, specialists, and families in their choice of scoliosis apps. **Conclusions:** The use of ICT solutions for spinal curvature assessment and monitoring brings several advantages to both patients and orthopedics specialists. Six scoliosis apps and one web-based application are evaluated, and a guideline for their selection is provided.

## 1. Introduction

Scoliosis comes from the Greek Word “skoliosis”, meaning crookedness [1]. Scoliosis is defined as a three-dimensional deviation of the normal vertical line of the spine. Spinal deformities include changes in the shape and position of the spine, thorax, and trunk in the coronal and sagittal planes as well as axial rotation of the vertebrae. Several different types of scoliosis exist, and the causes of scoliosis can differ.

The most prevalent types are idiopathic, congenital, and neuromuscular scoliosis and scoliosis in adults. A common form of scoliosis, which occurs without an obvious cause, is idiopathic scoliosis (IS). There is no identifiable and clear agent causing the deformity in IS. The individuals with hormonal problems, asymmetric growth, and who have a family member with scoliosis might suffer from IS.

Depending on the age at which it is first detected, IS can be classified as infantile (age 0–3), juvenile (age 4–9), or adolescent (age 10 up to maturity). There is a fourth category of scoliosis, known as adult idiopathic scoliosis, which is a continuation of the disease from childhood [2].

IS is extremely rare in infancy and early childhood. It spontaneously resolves in most cases but affects 2–3% of adolescents. Adolescent idiopathic scoliosis (AIS) occurs in individuals between the ages of 10 to 18, and it is associated with the faster progression of spinal curvature. When it is diagnosed, tt is important to evaluate the patient age and the magnitude of the curve to determine whether the curvature will progress and create potential long-term complications. Indeed, an accurate evaluation of the patient’s growth rate and the risk of curve progression is crucial to undertake the right interventions in a timely manner and only for patients who really need it [1].

The majority of AIS cases develop in females, and they cause the body to tilt to the right. Adolescent idiopathic scoliosis rarely causes pain. The severity and progress of spinal deformity is usually assessed by measuring the Cobb angle on spinal X-ray radiographs [3]. Cobb angle measurement is important for selecting treatment strategies and methods and for evaluating therapeutic effects in AIS. The Cobb angle is measured by determining the most tilted vertebrae on the superior and inferior aspects of the apex.

The lines are drawn along the top of the superior tilted vertebra and the bottom of the inferior tilted vertebra. Two additional lines are then drawn perpendicularly to those. The angle of intersection of the perpendicular lines is the Cobb angle expressed in degrees (see Figure 1) [4,5]. The Cobb angle represents the gold standard in the quantification and evaluation of scoliosis.

To be considered scoliosis, the curvature of the spine in the frontal plane must be at least 10 degrees. Curvature under 10 degrees is considered a normal variation. Curvatures between 10–20 degrees are considered as mild scoliosis.

Scoliosis severity is moderate when the Cobb angle ranges from 20 to 40 degrees. The condition of scoliosis becomes severe when the Cobb angle exceeds 40 degrees. As the Cobb angle increases, the severity of the curve increases.

Cobb angle plays an important role not only in diagnosis but also in the follow-up treatment because it can be used as an indicator to follow the evolution of scoliosis during therapy.

There are different procedures to determine the Cobb angle, and these are: manual procedure (traditional methods), digital computer-assisted (semi-automatic) procedure, automatic procedure, and smartphone app procedure, as will be explained in more detail in the next paragraphs. The deformity of spine and thoracic cage caused by scoliosis affects the appearance of individuals and can negatively impact on their psychological wellbeing.

In severe cases, scoliosis also leads to irreversible impairment of lung function and respiratory failure.

The rapid development of technology provides new ICT solutions (i.e., apps and web applications) that allow performing quick and cheap diagnosis and evaluation of scoliosis, ensuring an accuracy similar to the traditional methods.

The COVID-19 pandemic has had a heavy impact on public healthcare systems. In particular, access to health services has been compromised and, as a result, delays in the diagnosis of scoliosis frequently occur [6]. Delayed diagnosis of scoliosis can increase the patient’s risk of requiring surgical intervention.

The spread of scoliosis screening systems that use apps and web-based applications can facilitate the early detection of scoliosis on a global scale, mitigating the adverse impact of the COVID-19 pandemic on the diagnosis and management of scoliosis. The increased availability of such apps and the great pressure to use ICT-based solutions at home, due to COVID-19, are the main reasons to provide guidelines for doctors and patients in the adoption of apps.

Thus, the main motivation for this work is to provide a guideline for the choice of scoliosis management apps that considers both doctors’ needs (e.g., which angle the app measures, if the doctor can prescribe physical exercises at home, etc.), and patients’ needs (e.g., easiness of use, availability on app store, eventual fee to pay, etc.). Some related works are described below. For instance, the purpose of the study in [7] is to investigate whether the available apps for measuring the scoliosis grade are able to appropriately reproduce measurements taken from a traditional scoliometer. The authors in [8,9] evaluate the reliability of some apps for Cobb angle measurement. The study in [10] demonstrates the validity and reliability of an app for assessing deformity in scoliosis.

However, to the best of our knowledge, no study proposes a classification and evaluation methodology to assess the main features of the available scoliosis ICT solutions. Likewise, no study provides a guideline to support doctors or patients in their choice of scoliosis tools.

In order to fill this gap, the contribution of this paper is manifold: first, it proposes a methodology for the systematic assessment of ICT-based scoliosis tools; second, it provides the researchers with an evaluation of the main technical and functional features of the most popular apps and web-based applications for scoliosis; third, it provides an easy to use guideline that aims to help doctors, specialists, and those who wish to perform scoliosis checks at home in their selection of ICT-based scoliosis tools, considering several underlying aspects of interest for medical professionals and patients.

The rest of the paper is organized as follows. Section 2 describes the traditional, often manual, methods for scoliosis management and for taking measurements. Section 3 surveys the main ICT-based methods, with focus on web applications and smartphone apps. Section 4 introduces a methodology for evaluation of the scoliosis apps and web applications.

Section 5 presents the results for assessment of the scoliosis ICT solutions obtained through the application of the methodology described in the previous section. Section 6 discusses the assessment results and provides some guidelines to help users in choosing an ICT-based scoliosis tool. Finally, Section 7 concludes the paper and underlines future work.

## 2. Traditional Scoliosis Management

Traditional scoliosis management comprises two main activities: (i) the assessment of spinal curvature through radiological imaging; and (ii) the physical examination of the patient, in which a goniometer (scoliometer) is employed to follow and estimate the angular deformity of the spine.

### 2.1. Radiography Analysis

Recent advances in imaging technology have resulted in a better understanding of scoliosis and its progression. Radiography remains the common method for the evaluation of treatment efficacy as well as the monitoring of disease progression. The most common and effective technique for measuring spinal curvature on radiographs is still Cobb angle measurement.

Traditionally, Cobb angle measurement is performed on radiographic images using a pencil and a goniometer. In particular, in the manual procedure based on radiological analysis, the lines are manually drawn onto a radiographic film of the spine, and the angle formed by the most inclined vertebrae is determined by using a protractor. The manual procedure is simple but has disadvantages, because it can be time-consuming and has a low accuracy and poor repeatability due to human subjectivity and significant inter-observer and intra-observer variations found in the value of Cobb angle [11].

The use of different protractors, markers/pencils of varying width, poor quality images, and the level of experience of the observer are also intrinsic causes of error [12]. However, if the Cobb angle is to be used clinically, for example to guide a treatment plan, it is important that its estimation is highly reliable [13]. Systems that measure the Cobb angle on digital radiographs are described later.

### 2.2. Manual Measurement with Scoliometer

Radiological examination is the conventional modality to investigate scoliosis. However, X-ray imaging is strongly related to cancer occurrence due to several radiographic examinations in which the patients are subjected during childhood and adolescence [14]. Because children are more radiosensitive than adults, tools based on tablet PCs or smartphones can limit the radiation exposure and thus limit the incidence of such cancers [15,16].

A simple and non-invasive evaluation tool is a physical examination. This physical examination implies the Adam’s forward bend test [17]. The Adam’s forward bend test is a test used at schools and doctors’ offices to check for scoliosis. The exam is named after English physician William Adams, who described it in 1865.

The use of a spine-measuring scale or scoliometer is combined with the Adams forward bend test [18]. The scoliometer, placed over the spine while a person is in a forward-bending position, measures the angle of trunk rotation (ATR) [19]. The scoliometer is characterized by a metal sphere inside a water recipient that dislocates on a range of 0–30° for both sides and indicates the angle of axial trunk rotation. There is a strong relationship between the back asymmetry measured using a scoliometer and the Cobb angle measured by radiography.

Then, the scoliometer can replace X-rays in screening or follow-up of patients with idiopathic scoliosis. This has great potential, as it can reduce medical costs and allow avoiding recourse to measurements using ionizing radiation. In addition, the scoliometer is a faster and more easily accessible clinical tool, especially in those areas where X-rays are not available [20]. However, the scoliometer is not usually used as a diagnostic tool, and ATR values are used to identify the patients who need more careful clinical evaluation [21].

## 3. ICT-Based Scoliosis Management

The enhancement in ICT technology as well as the increasing use of smartphone devices among the population have led to great benefits in the health sector. Moreover, the widespread use and ubiquity of information and communication technology have encouraged the shift from a doctor-centered to a patient-centered approach [22]. In this context, several studies show that the ICT solutions can be effectively used in the clinical practice, and they can replace those methods based on the traditional goniometer technique [23,24,25].

In particular, smartphone-aided measurement of Cobb angle or ATR shows high reliability and efficiency. ICT-based scoliosis management includes two approaches: (i) standalone software tools or web-based applications; and (ii) smartphone applications, commonly named apps. In ICT-based scoliosis management, semi-automatic or fully automatic measurement procedures are usually used.

In the semi-automatic procedure, landmarks are placed on images of the patient using a computer mouse, and the angle is then automatically determined. This procedure has proven reliable and to have less variation than the manual procedure, although this depends on the experience of the observer [12].

It has been noted that the reliability and consistency of Cobb angle measurement improve with the help of computerized methods. Digitalized X-ray images are manipulated using software tools installed on a personal computer; these tools enable modifications of the brightness, contrast, and magnification of images, thus leading to better visualization of details and more consistent results [12].

Artificial intelligence (AI) applications can be used to streamline clinicians’ workflow, improve efficiency, and reduce errors. A deep neural network (DNN) can predict vertebral slopes and automatically calculate Cobb angle after the system has been properly trained with enough data. However, the method is not fully automatic, as the user has to assign vertebral patches in posteroanterior (PA) radiographs. A rectangle of 150×150 pixels is created at each patch and sent to the DNN to obtain the slope. The Cobb angle is then automatically calculated as the sum of the maximum absolute slopes with respect to the horizontal line [26]. This method has the potential to reduce the variations between one measurement and another, improving the repeatability of Cobb angle measurements and allowing reliable and objective assessments of scoliosis.

A fully automatic method may be useful to reduce or completely eliminate the variability related to user interaction, such as the one presented by Forsberg et al. (2013) [27]. However, access to this technology is not universal, and computer-aided measurement techniques are essentially used for research purposes [28]. For example, there are geographical areas from which physicians are unable to access digital tools. In these cases, smartphones offer a convenient tool to directly measure the Cobb angle either on digital radiographs or from a printed version of the radiograph [28].

With the increasing use of smartphone devices, software programs for cell phones, known as applications or apps, can be integrated into clinical practice and replicate the function of traditional medical devices, assisting doctors and spinal surgeons in the calculation and measurement of spinal curvature angles [10].

The angle is determined by aligning the smartphone to the spine, and it is then displayed on the smartphone screen [11]. It has been demonstrated that measurement of the Cobb angle using a smartphone is equivalent to that using the traditional protractor [28]. Smartphone app represents an effective screening option particularly in the low- and middle-income countries, where the lack of physical clinical scoliometer devices may lead to many cases of undetected scoliosis among children, with serious consequences. Smartphones contain accelerometer, gyroscope, and GPS sensors, which allow reading of information. Smartphone applications can be useful for allowing continuous and easy tracking and monitoring of scoliosis. Patients can monitor their curvature progress from home on their own and obtain clinical consultation without an in-person visit [29].

Furthermore, there are apps that, when storing previous measurements, allow comparison of current readings with previous readings for a particular patient [28].

Data coming from apps can also be used for scientific research analysis and help in formulating trends and treatments [15]. Smartphone applications have the advantage of being highly stable, highly precise, and providing quick assistance in the diagnosis and treatment of the disease.

Apps can improve the accuracy of screening by physicians and be used to detect scoliosis in patients on a global scale [30]. Figure 2 shows the reference architecture and graphical workflow of ICT-based scoliosis management. The graphical workflow shows the communication between doctor and patient through the app. In particular, we can identify three software layers in the scoliosis management workflow:**Patient layer**: The patient periodically takes scoliosis measurements at home through the app. The sensors or the camera of the device on which the app is installed allow data to be collected.**Data layer**: Data coming from patients are stored on the back-end database web portal and associated with the patient’s medical record for management of the individual patient and, after aggregation, for machine learning. In fact, the use of apps on a large scale allows the collection of big data, which may be used to train machine learning algorithms to produce predictive models, e.g., time to recovery, follow-up, clinical outcome, etc.**Medical layer**: The doctor, through the use of an app or web application, accesses the data relating to scoliosis measurements and other information of patients and can perform several functions, such as (i) sending, to the patient, some exercise prescription to be performed at home; (ii) monitoring the efficacy of treatment; (iii) giving immediate feedback to the patient without waiting for a visit to the studio; (iv) sending appointment reminders to the patient.

### 3.1. Web-Based Applications

Web-based applications for scoliosis management have the advantage that they do not require specialized software—only a web browser is used. On the other hand, for typical measurements related to scoliosis diagnosis and monitoring, specialized devices and sensors are needed that are unavailable on a typical browser. For this reason, web-based applications are a useful tool for clinicians that already own all the sensors and devices necessary for taking measurements and that can load such measurements into the web application. On the other hand, web applications are easy to use for patients, who can access prescriptions and exercise in an easy way.

#### Scoliosis Manager

Scoliosis Manager (https://www.scoliosismanager.it/, accessed on 10 January 2023) is a web-based system for scoliosis management that is available in more than 10 languages. The system has been developed by the Spine Italian Scientific Institute (Istituto Scientifico Italiano Colonna Vertebrale—ISICO) (ISICO is one of the leading institutes in the world in the treatment of vertebral deformities, https://www.isico.it/, accessed on 10 January 2023) and provides a free internet-based portal that may be used by clinicians for managing scoliosis rehabilitation. Medical doctors [31] and physiotherapists [32] are the primary users of the system and, as a preliminary step, they need to register to the system and can in turn register their own patients. Clinicians can use the system for monitoring rehabilitation steps, including assigning exercises and monitoring their execution. The exercise menu offers a very rich set of exercises that may be assigned to the patients. The main functions offered by the system are: (i) searching for patients in the system; (ii) measurement of pathological vertebral rotation on radiographs using the Raimondi method (axial vertebral rotation (AVR) is an important measure in scoliosis assessment and is defined as the rotation of a vertebra around its longitudinal axis when projected onto the transverse image plane. The main methods for assessing the AVR on radiography are the Raimondi method, using the so-called Raimondi’s tables, and the Perdriolle method, recommended by the Scoliosis Research Society [33]); (iii) storage of temperature values measured by certain kinds of orthopedic corsets for scoliosis (e.g., the ISICO corset equipped with a ThermoCare temperature sensor). An interesting feature of Scoliosis Manager is the so-called Virtual Patient, which may be used by the clinicians to simulate the execution of visits, assignment of exercises, compilation of the clinical diary, and evaluation of such a virtual patient.

### 3.2. Smartphone Applications (Apps)

Due to the spread of the COVID-19 pandemic, ICT has been utilized to a much larger extent to limit contact between persons. Therefore, a greater number of people have become familiar with ICT technology, reporting high levels of satisfaction [34]. In this context, the smartphone has become the most widely used ICT-based system in the world, and it has been gradually introduced into clinical practice. Indeed, smartphones are able to provide healthcare services anytime and anywhere in a cheap way. Indeed, smartphones can replicate the function of traditional medical devices such as the scoliometer, saving cost and time (e.g., for stable cases, where the period between follow-ups can be lengthened), thus improving clinical efficiency and convenience [10]. On the other hand, smartphone apps may overcome those limitations.

In recent years, a plethora of ICT-based solutions for healthcare have been developed, and the development of new ICT tools and methods is ongoing, including of apps and web-based applications. To select the apps to evaluate, we performed an extensive search both by analyzing the literature discussing scoliosis management through apps [35] and by consulting the web. Moreover, some apps were suggested by the orthopedist participating in this work. Some apps that were cited in the literature but unavailable on the store were excluded. Thus, we are confident that the systems discussed in the following are the most popular apps for scoliosis management.

#### 3.2.1. ScolioTrack

ScolioTrack allows the patient to track, from month to month and between one visit and another, the progression of the spinal curve by using the iPhone accelerometer. This application has a high degree of accuracy, which makes it suitable for use by professionals for monitoring of multiple patients. Furthermore, it is simple enough for personal use at home and allows storage of scoliosis measurements for monitoring. ScolioTrack measures the patient’s ATR and also keeps track of height, weight, age, and photo records of the person’s back to allow later comparisons and follow-up [15] (see Figure 3). A display shows measurements in graph format for easy monitoring of changes in scoliosis over time. Among the various functions, the app allows taking a photo of the person’s back through a camera and saving the data of multiple users. "Grid view mode" allows users to easily notice any visual changes to the back, such as rib humps, hip protrusion, body alignment, or spinal deviation, and easy comparison with previous photo records (https://scoliotrack.com/, accessed on 10 January 2023). The app offers answers to some of the frequently asked questions about scoliosis, and it displays the latest news feed on the topic to keep users up-to-date. ScolioTrack is proven to be a safer, simpler, and cheaper method than radiological checks.

#### 3.2.2. Scoliometer

Scoliometer is a useful app for professionals and those who want to monitor changes in the spinal curvature at home. The app, at a charge, represents a simple and fast way to accurately measure and monitor the ATR (see Figure 4). Furthermore, Scoliometer can screen scoliosis curves up to 50 degrees, and it is compatible with the latest iPhone and Android devices (https://apps.apple.com/gb/app/scoliometer/id649380964?l=it, accessed on 10 January 2023). Scoliometer can be considered a simpler and cheaper version of the ScolioTrack app [15].

#### 3.2.3. APECS

The APECS (Artificial Intelligence Posture Evaluation and Correction System) app takes advantage of artificial intelligence for the evaluation and correction of the patient’s posture. It allows tracking the patient’s posture and gives daily tips on how to improve and maintain good posture as well as feedback on the posture exercises done at home. Specifically, APECS performs the posture evaluations using markers that are positioned on the patient body photo, and it uses photogrammetric algorithms for performing accurate body symmetry assessment. The user can choose between the auto-detection or manual positioning of the markers following the guide. APECS includes the exercises module by which the doctor can prescribe a specific workout on the basis of the posture assessment and the patient needs to lead to positive results in correction (https://apps.apple.com/it/app/apecs-ai-posture-evaluation/id1488264106, accessed on 10 January 2023) (see Figure 5).

#### 3.2.4. CobbMeter

CobbMeter is used to measure the Cobb angle, the kyphosis angle, and the sacral slope on vertical spine through the radiographs. This method for measuring the Cobb angle is a simple and effective procedure widely used in clinical practice. The CobbMeter app can facilitate this measurement with excellent reliability and efficiency [36]. CobbMeter is an app designed for spinal professionals, and it turns the iPhone into a professional measuring tool. The app is freely available, and authorization by the developers is not required. The CobbMeter app calculates the Cobb angle on radiographs using the angle sensor available on the iPhone (microelectromechanical system accelerometer—MEMS). First, the upper side of iPhone is aligned to upper vertebra, and the position is validated by clicking on the screen.

Next, the lower side of the iPhone is aligned with the lower vertebra, and it is validated again. The Cobb angle is then measured by the sensor used by the app, and it is shown on the iPhone screen. Measures are recorded and can be sent by e-mail to the surgeon. A calibration procedure is implemented to accurately measure the angle with respect to the horizontal plane as in the sacral slope.

The use of the CobbMeter app is considered safe and applicable for clinical practice [8] (see Figure 6).

#### 3.2.5. ScolioDetector

ScolioDetector is designed for parents, teachers, and doctors to measure the child’s spinal curvature using a built-in scoliometer function (see Figure 7). Moreover, it records the measurements readings for monitoring. The app is free of charge, and users can periodically self-screen. Measurements of more than five degrees may indicate early signs of scoliosis, and patients should seek for professional medical advice. ScolioDetector is a program widely used by people all over the world.

#### 3.2.6. Scoliosis Tracker

Scoliosis Tracker is designed to measure, record, and track adolescent’s scoliosis. The Scoliosis Tracker app provides a digital scoliometer to measure spine rotation and determine the need for X-rays exams. It provides a checklist to track patient compliance with conservative scoliosis care (e.g., bracing, vitamin D, and physical therapy) and also includes appointment reminders and several examples of educational content about scoliosis (https://apps.apple.com/it/app/scoliosis-tracker/id1464868041, accessed on 10 January 2023) (see Figure 8).

## 4. A Novel Methodology for Comparing Scoliosis Apps and Web Applications

In this section, we present the methodology used for assessing and comparing the scoliosis web applications and apps described in Section 3.1 and Section 3.2, respectively (they will hereafter be referred as ‘tools’ for simplification purposes). The methodology, based on qualitative evaluation approach, defines 5 aspects, named macro-categories, to assess the tools. Each macro-category includes a group of homogeneous categories defined to rate more in detail the tools functionalities and features. The macro-categories and categories are chosen with the aim of covering the main aspects that are common and meaningful for all scoliosis tools. The 5 macro-categories are: Availability, Technology, Measurement, Functions, Qualitative evaluation. Macro-categories and categories are defined below.

−The *Availability* macro-category includes the three categories:•*Reachability*, which indicates the source from which the app can be downloaded (i.e., App Store and/or Play Store) or the URL through which the web-based application is reachable;•*Accessibility*, which provides information on the payment (i.e., if the tool requires payment of a fee for its download/use or if it is freely downloadable/usable);•*User Support*, which specifies whether a tutorial or a user guide is provided. For instance, some payment tools provide a demo version with limited functionalities to introduce the users to the full tool interface and key features.−The *Technology* macro-category includes the following two categories that provide information on the technological aspects of the tool:•The *Operating System* category reports the list of operating systems (OS) that are compatible with the tool (i.e., Apple OS and/or Android OS for apps and Windows and/or Linux for web applications);•The *Additional Devices* category indicates if special equipment is required by the app. More specifically, some apps require a coupled plastic accessory that has a groove to allow it to be placed under the smartphone device frame and on the top of the patient back. This accessory allows better following the curvature of the back and thereby improves the precision of measurement.−The *Measurement* macro-category provides information on the measurements used for assessing the grade of the patient’s spinal deformity. It includes three categories:•The *Sensors* category specifies the type of smartphone-embedded sensors (i.e., accelerometer, gyroscope, and camera) that are used to acquire the measurements;•The *Type of Measurement* category indicates the method (i.e., ATR, Cobb angle) used to assess the severity of the spinal deformity;•The *Measurement Results Handling* category indicates the format in which the measurement results are presented and how they are handled. For instances, if the measurement is tracked over time, if symmetry test to assess the scoliosis grade is performed, and so on.−The *Functions* macro-category provides information about the functionalities of the tool. It includes four categories:•*Posture Monitoring*, which specifies if the app monitors patient posture over time. The apps that provide this functionality are able to send real-time sounds and visual notification in the case of bad posture detection;•*Dedicated Medical User Interface* specifies if a doctor-dedicated interface is available. Some tools have different kind of interfaces depending on the type of user, i.e., patient or doctor. For instance, when the doctor accesses the app with their own credentials, some functionalities, not provided for the patient accounts, are enabled (e.g., tracking of patient records or appointment reminders);•*Patient Prescriptions and Exercise Monitoring* specifies if the tool provides further functionalities of workout monitoring or, in addition, if it provides prescriptions for exercises that patients should undertake each day at home. These functionalities are often useful for encouraging adherence to physical exercise;•*Additional Patient Information* specifies if the tool uses additional patient information, such as age, weight, height, etc., for scoliosis classification and monitoring. Depending on its classification, beside of the patient weight and height, the scoliosis problem is managed differently.−The *Qualitative evaluation* macro-category reports the results for the evaluation of the tools based on a qualitative approach. It includes two categories:•The *Strengths* and *Weaknesses* categories that respectively show the strengths and the weaknesses of the tools;•The *User-friendliness* category is related to the levels of computer competency required to use each tool. If low computer skills are required (basic computer knowledge) then user-friendliness is high. If medium computer skills are required (limited computer knowledge) then user-friendliness is medium. If high computer skills are required (expert to manage software package, module, and/or programming code) then user-friendliness is low.

The main rationale for the definition of an evaluation methodology is to have a methodological framework that can be used not only in this work to evaluate the current apps but also in future works when new apps are available. As previously described, the proposed methodology for the evaluation of scoliosis apps considers five macro-categories that are relevant either to the patients (e.g., the availability or the cost, the possibility to receive prescriptions of exercises at home), or to the clinician (e.g., some technical features, such as the angles that the app can measure or the possibility to remotely monitor the patients or to prescribe them exercises at home). Moreover, considering that several apps for scoliosis management have been developed and are publicly available and that choosing which app is the right fit for the user can be a time-consuming task and lead to confusion, we also provide a simple guideline for choosing the apps. Having the previous methodology helps in compiling the guideline.

As previously described, the five macro-categories consider: (i) technological aspects (e.g., available sensors, how angles are measured); (ii) the type of measurements (e.g., Cobb angle, angle of trunk rotation, axial vertebral rotation); (iii) availability (e.g., app store and eventual fee to pay); (iv) the functions offered to the user (e.g., posture monitoring, exercise prescription); (v) an overall evaluation (e.g., pros and cons, usability).

Such categories come from our experience in the analysis of other ICT-based tools and apps in the bioinformatics [37] and biomedical [38] fields. Although those surveys have investigated tools in different domains with respect to scoliosis management, those tools share, with scoliosis management apps, the fact that the user is not an ICT expert (e.g., clinicians or bioinformaticians) and the fact that they manage biomedical and clinical data.

Moreover, we are aware of the System Usability Scale (SUS), which is a methodology for measuring the usability (i.e., the subjective perception of interaction between the user and the system) of a wide variety of products and services, including hardware, software, mobile devices, websites, and applications.

Although the SUS has many benefits that make it popular for the usability assessment, it has many limits. For instance, SUS primarily measures the perceived usability of a system rather than the overall system usability [39,40]. Moreover, although SUS covers several aspects related to the usability, such as complexity and learnability, it is however unable to provide accurate information on the system weakness.

Moreover, recent studies about SUS have reported that users having a more extensive experience with a system tended to provide higher and more favorable evaluation with respect to users with no or limited experience with the system (i.e., it can be affected by bias). Furthermore, SUS omits important aspects that are specific for the type of system (e.g., fashion and ergonomic design for wearable healthcare tools).

Our proposed methodology based on the five categories goes beyond the SUS usability measurement to provide an overall assessment of ICT scoliosis tools. Indeed, it covers technological aspects (e.g., operating system, additional devices), measurement (e.g., type of measurement), qualitative evaluation (e.g., strengths and weaknesses) that are not included in SUS evaluation.

## 5. Comparison of the Scoliosis Apps and Web Applications

The methodology illustrated in Section 4 is used to compare the tools described in Section 3.1 and Section 3.2. Table 1, Table 2, Table 3 and Table 4 (all four tables will hereafter be referred to as ‘comparative tables’) show the results of the assessment in a easy to understand and intuitive way that can help doctors, specialists, and all those who wish to perform scoliosis checks at home in their choice of ICT-based tool.

Table 1 shows the comparison of the six assessed apps on the basis of the availability and technology macro-categories. In particular, considering Table 1, half of the assessed apps are supported by both iOS and Android OS, and they provide guide and/or video tutorial. Availability on both Apple and Android platforms makes the app reachable by more users. Three apps are available in both Apple and Play Stores, two only on Apple Store, and one only on Play Store. None of the assessed apps requires additional devices for their use.

With regard to Scoliosis Manager, it can be accessed via the Internet and is free of charge. Moreover, its usage does not require the installation of local software.

Table 2 shows comparison of the six assessed apps on the basis of the measurement macro-category.

Considering Table 2, we can divide the scoliosis apps into two categories, i.e., sensor-based and camera-based, depending on the technology type that is used for acquiring the measurements. Sensor-based apps use smartphone-embedded motion sensors, such as accelerometers and gyroscopes. The accelerometer is used to detect the linear acceleration of the device, while the gyroscope measures its angular velocity expressed in degrees. Both are used in combination for detecting the curvature variation of the spine in the sagittal and coronal planes [41].

Camera-based apps use the standard two-dimensional (2D) digital cameras of the smartphone to measure the ATR or the Cobb angle. An example of a camera-based app is APECS, which uses the anthropometric landmarks and anatomical angles for the postural assessment.

In general, although the image-based systems have proven their potential in many contexts and, in particular, contributed to the development of several scoliosis detection systems [42,43], we must consider that the measurements acquired by the image-based systems are affected by several variables, such as the camera resolution or the lighting conditions, which can lead to inaccuracy in the measurements. Electronic sensors, such as accelerometers and gyroscopes, are widely applied by apps to address the weaknesses in the image-based systems [44]. Nonetheless, other problems regarding the use of the sensor-based apps have emerged, such as, for example, the disturbance due to the trunk movement and the gyroscope drift (environmental stress and temperature effects are major sources of drift).

**Table 2 ijerph-20-05520-t002:** Comparison of the tools with respect to the measurement macro-category. Note that only APECS uses the camera to acquire measurements. APECS draws custom angles on the patient photo to perform the Adam’s forward bend test assessment [45].

	Measurement		
Name	Sensors	Type of Measurement (ATR, Cobb, Other)	Measurement Results Handling
ScolioTrack	Accelerometer Gyroscope	ATR	- Keeps track of the patient’s ATR measurements; - Tracks the patient’s height and weight and records photos of the person’s back; - Shows the patient back curvature progression in a graphical format.
ScolioMeter	Accelerometer Gyroscope	ATR	Measures the patient ATR.
APECS	Camera	Custom angles on photos	- Locates markers on the patient body photo for: (i) drawing angles between the lines connecting the markers; (ii) performing symmetry tests and evaluations on-distance; (iii) performing the bend test assessment. - The resulting measurements are used to evaluate the posture and scoliosis grade, and they are provided as a pdf report in a tabular or graphical way to be saved and shared.
CobbMeter	Accelerometer Gyroscope	Cobb angle Kyphosis angle Sacral slope	- Uses regular radiographs or even those on the computer screen to measure the angles. The measurements can be saved on smartphone.
ScolioDetector	AccelerometerGyroscope	ATR	- Measures and records the patient ATR.
Scoliosis Tracker	Accelerometer Gyroscope	ATR	- Measures and records the patient ATR - Tracks curve progression over time.
Scoliosis Manager	Raimondi’s method	AVR	- Manages all clinical activities related to the treatment of spinal deformities: medical examination, rehabilitation evaluation, exercise plans.

**Table 3 ijerph-20-05520-t003:** Comparison of the tools with respect to the functions macro-category. Note that only CobbMeter has a dedicated medical user interface because it is intended for the spinal care professional. “-” means the information is not available.

	Functions			
Name	Posture Monitoring	Dedicated Medical User Interface	Patient Prescriptions and Exercise Monitoring	Additional Patient Information (e.g., Age, Weight, Height)
ScolioTrack	-	-	-	Age, weight, height
ScolioMeter	-	None	-	-
APECS	Yes	None	Yes	-
CobbMeter	None	Yes	-	-
ScolioDetector	None	None	Yes	Age, height, weight, body mass index (BMI)
Scoliosis Tracker	None	-	-	Height
Scoliosis Manager	-	Yes	Yes	Temperature

**Table 4 ijerph-20-05520-t004:** Comparison of the tools with respect to Qualitative Evaluation macro-category. Note that all the tools have a high user-friendliness and their weaknesses are due to the type of the used sensors (i.e., camera-based or sensor-based).

	Qualitative Evaluation		
Name	Strengths	Weaknesses	User-Friendliness
ScolioTrack	- shows data output in graph format; - tracks changes over time; - saves and tracks the data of multiple users.	- can be affected by biases due to the gyroscope drift	Medium
ScolioMeter	- easy to use.	- no more functionalities in addition to that of scoliometer; - can be affected by biases due to the gyroscope drift.	High
APECS	- saves, exports and shares results; - receives textual interpretation of the results; - shows results in a graphical way; - provides daily tips for posture improvement; - indicates best exercises for posture correction, muscles and core strengthening, pain relief.	- measures reliability is affected by the camera resolution and the lighting conditions.	Low
CobbMeter	- clinical practice improvement.	- only measurements on the radiography are available.	High
ScolioDetector	- records the readings; - scoliosis exercises section is available.	- can be affected by biases due to the gyroscope drift.	Medium
Scoliosis Tracker	- checklist to track the patient compliance with conservative scoliosis care; - appointments reminder; - educational contents about scoliosis.	- can be affected by biases due to the gyroscope drift.	Medium
Scoliosis Manager	- software works over the internet; - no software to install on your own PC; - access to data from any location connected to the internet; - in the exercises area of the menu it is possible to find a very rich set of exercises.	- it is recommended for only trained personnel.	Medium

Examples of sensor-based apps are *ScolioTrack, ScolioDetector, CobbMeter*, and *Scoliosis Tracker*. They are able to record additional patient information such as height, weight, age, and body mass index (BMI). Except for CobbMeter, which measures the Cobb angle, the kyphosis angle, and the sacral slope on vertical spine radiographs, the other five apps measure the ATR. Unlike the other tools, Scoliosis Manager is a web application, and it uses a different type of parameter and method to evaluate the severity and predict the progression of scoliosis, i.e., the AVR parameter and the Raimondi method.

Table 3 shows a comparison of the six assessed apps on the basis of the functions macro-category. Considering Table 3, we can find the main functions provided by the apps. Only one app does posture monitoring. APECS and ScolioDetector make exercise prescriptions and provide additional patient information such as age, height, and weight. Scoliosis Tracker provides height. Scoliosis Manager makes exercise prescriptions and provides the temperature values of some kind of orthopedic corsets. CobbMeter is the only app that has a dedicated medical user interface.

Table 4 shows comparison of the 6 assessed apps on the basis of the qualitative evaluation macro-category. Considering Table 4, apps are rated on their strengths, weaknesses, and user-friendliness. Scoliometer and Cobbmeter have a high user-friendliness. ScolioTrack, ScolioDetector, Scoliosis Tracker, and Scoliosis Manager have medium user-friendliness. APECS has low user-friendliness.

Each app has its strengths and weaknesses. The weaknesses of apps are due to the type of the used sensors (image-based or sensor-based). For example, ScolioTrack records the readings and shows them in graph format; a weakness is that the measurement can be affected by biases due to the gyroscope drift.

### Simple Guideline to Choose a Scoliosis App

Here we provide a guideline that, starting from the Table 1, Table 2, Table 3 and Table 4, can facilitate the choice of the ICT-based scoliosis tool. Firstly, the user should select only the tools supported by the OS that is installed in their own device. Subsequently, they can choose an app rather than another according to the functionalities provided by the app.

In particular, if the user only needs immediate feedback on the current state of scoliosis, it may be sufficient to choose a simple app that provides only the basic scoliometer functionality such as, for example, the *ScolioMeter* app. Instead, if the user needs further functionalities such as the one that records over time the scoliosis measurements to monitor the scoliosis progression, *ScolioTrack*, *Scoliosis Tracker*, and *ScolioDetector* could be good choices.

The availability of specific functions could be another valid option for selecting one scoliosis tool over another. For instance, the exercise prescriptions and monitoring functions can be useful to encourage exercise adherence, which is still a significant factor in determining the success of scoliosis treatment. In addition, the possibility to export and share the results in a tabular or graphical format (e.g., progression curve of scoliosis) can boost regular posture screening as well as patient–doctor engagement and communication.

## 6. Discussion

To summarize the analyzed apps in a chart, we take into consideration the user-friendliness category (see Table 4) and a combination of the functions macro-category (see Table 3) and Measurement Results Handling category (see Table 2). We define "Wealth of functionality" as a combination of two categories, “Functions” and “Measurement Results Handling”. In other words, “Wealth of functionality” comprises the “Functions” macro-category and the “Measurement Results Handling” category. The combination takes place by counting the functions contained in the Functions macro-category, those contained in the “Measurement Results Handling” category and adding them.

Figure 9 shows, through a two-dimensional chart, the results for the assessment of tools carried out in the previous sections. The chart focuses only on two dimensions among those described in Section 4, which are “Wealth of functionality” and “User-friendliness”. We selected these dimensions among the others because, once the user selects the tools compatible with their own OS, these two dimensions are the primary deciding factors in the choice of the scoliosis tool. The two-dimensional chart axes report the qualitative ratings scale whose values are “low”, “medium”, and “high”.

These values are assigned to the tools with respect to each dimension depending on the number of functions provided by the tool and the ease of use. The rating scale used in the “Wealth of functionality” evaluation is defined in the Table 5, while that used in the “User-friendliness” evaluation is related to the previously defined required computer skills (low, medium, high) (see Section 4), where each required computer skill corresponds to a certain scale of ease of use (low, medium, and high).

For example, APECS provides at least six functionalities and is thus characterized by high “Wealth of functionality” with respect to the others, although it has low “User-friendliness”. Indeed, the manual positioning of the markers could be not intuitive and, thus, could require the user to follow the guide. Note that ScolioMeter is the most easy to use tool among the others, but it provides only the scoliometer functionality. Thus, it is characterized by high “User-friendliness” and low “Wealth of functionality”.

Although CobbMeter provides few functionalities, it allows measuring, in a simple way, several type of angles (e.g., Cobb angle, kyphosis angle, sacral slope on vertical spine) through radiographs. Thus, it is characterized by medium “Wealth of functionality” and high “User-friendliness”. The rest of the tools are characterized by medium “Wealth of functionality” and medium “User-friendliness” in the area with respect to both dimensions.

## 7. Conclusions and Future Work

The use of smartphone apps and web applications for spinal curvature assessment and scoliosis progression monitoring brings several advantages to both patients and health professionals, in particular to orthopedics specialists. On the patient side, these tools can provide services for patients, such as: periodic monitoring of the progression of spinal curvature (from remote), taking measurements at home, and avoiding frequent in-person visits; in addition, patients can quickly obtain consultations and feedback on the correctness of the exercises and the efficacy of the therapy, thereby cutting down waiting times.

Such an ICT-based system promotes self-care management and strengthens patients’ involvement in monitoring their health conditions. On the medical side, on the other hand, these tools offer significant advantages such as reducing the workload of healthcare professionals and better management not only of the visit queue but also of the individual patient [1].

Doctors can manage a large number of scoliosis patients and give immediate feedback to the patient based on the comparison between the prescribed exercises and the results achieved up to that moment. Furthermore, ICT-based management allows simplifying inter-professional communication and exchange of information in a secure way, enabling doctors to remotely communicate the main health issues, overcoming logistic and long-distance criticalities.

Apps installed on mobile devices can help doctors to organize and track appointments, meetings, call schedules, and other clinical obligations [46].

The possibility of having a digital tool that is easy to use can encourage patients to self-screen and detect scoliosis early, preventing complications in the short and long terms. Moreover, the development of a scoliosis screening system that uses the standard two-dimensional (2D) digital cameras that come with tablet personal computers (PCs) and smartphones or sensors integrated in the smartphone will facilitate the efforts made to detect scoliosis patients on a global scale.

Finally, the measurements made by using apps are more stable and precise because they make use of sensors such as accelerometers and gyroscopes, which allows improving the accuracy of screening.

As the main contributions of this work, we have first defined a new methodology to evaluate the apps for the scoliosis management, then we prepared a summary of the six main apps for scoliosis management, and finally provided a guide on how a potential user can choose one of these apps.

A limitation of the proposed methodology is related to the number of assessed characteristics: although we tried to cover many characteristics of scoliosis management apps, it is possible that some important characteristics have been overlooked. In future work, we intend to overcome this limitation by extending the number of aspects and/or categories.

Another limitation of this work is related to the apparently low number of tools assessed in the study. In future work, we intend to overcome this limitation by extending on the number of the ICT-based scoliosis tools that will be compared. Furthermore, we plan to develop an app that brings additional functionalities with respect to current apps on the basis of the requirements provided by the orthopedic participating in this study.

Finally, the contribution of this paper to existing research and practice is manifold: first, a methodology for the systematic assessment of ICT-based scoliosis tools is proposed; second, it provides researchers with an evaluation of the main technical and functional features of the most popular apps and web-based applications for scoliosis; third, it provides an easy to use guideline that aims to help doctors, specialists, and those who wish to perform scoliosis checks at home in their choice of ICT-based scoliosis tools, considering several underlying aspects of interest for medical professionals and patients.

In conclusion, the use of apps in scoliosis management that allow the extensive collections of clinical data, beside easing clinical practice, opens the doors to the future application of machine learning and artificial intelligence in orthopedics.

## Figures and Tables

**Figure 1 ijerph-20-05520-f001:**
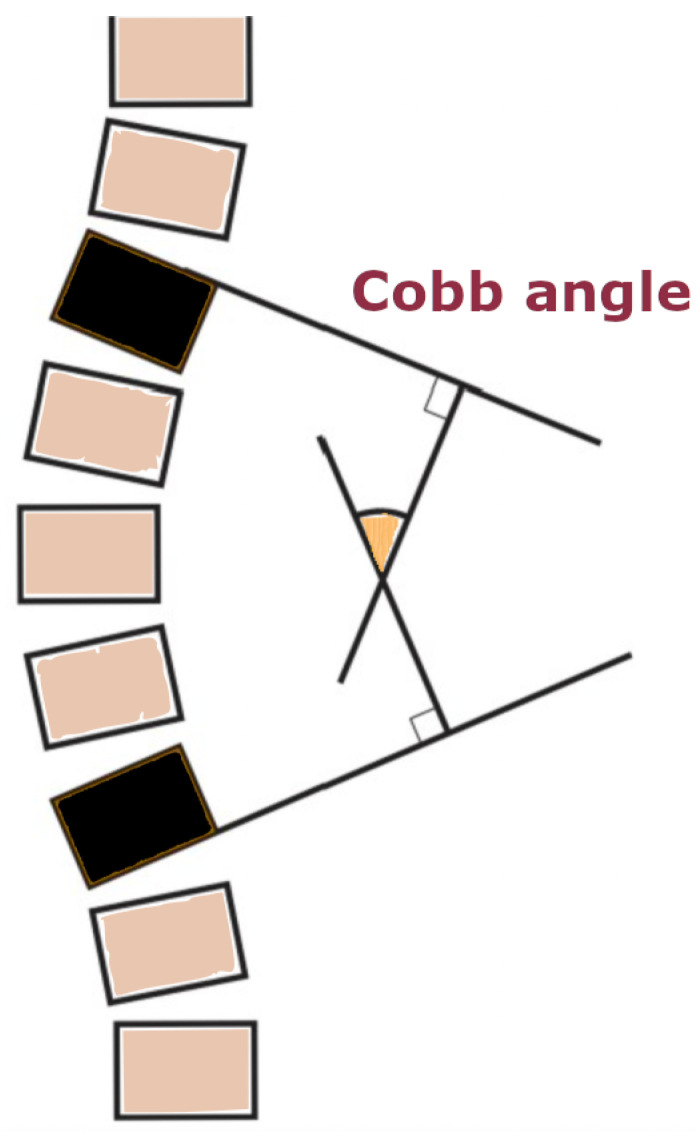
Quantification of scoliosis grade using the Cobb angle.

**Figure 2 ijerph-20-05520-f002:**
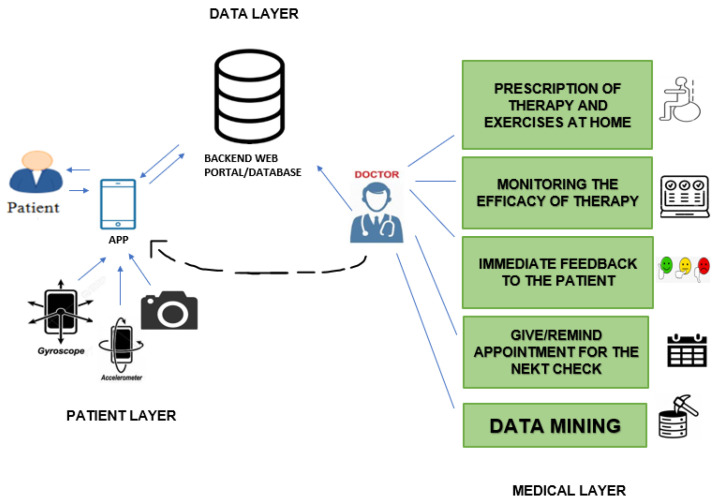
Workflow and reference architecture of ICT-based scoliosis management solutions.

**Figure 3 ijerph-20-05520-f003:**
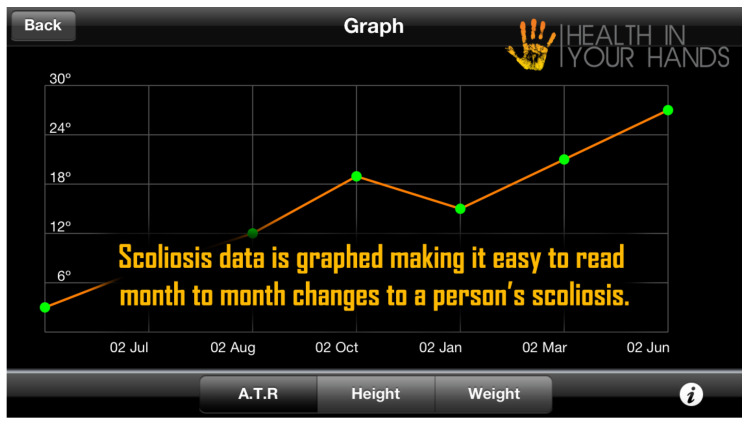
ScolioTrack. (This figure is a snapshot taken by the screen of the app).

**Figure 4 ijerph-20-05520-f004:**
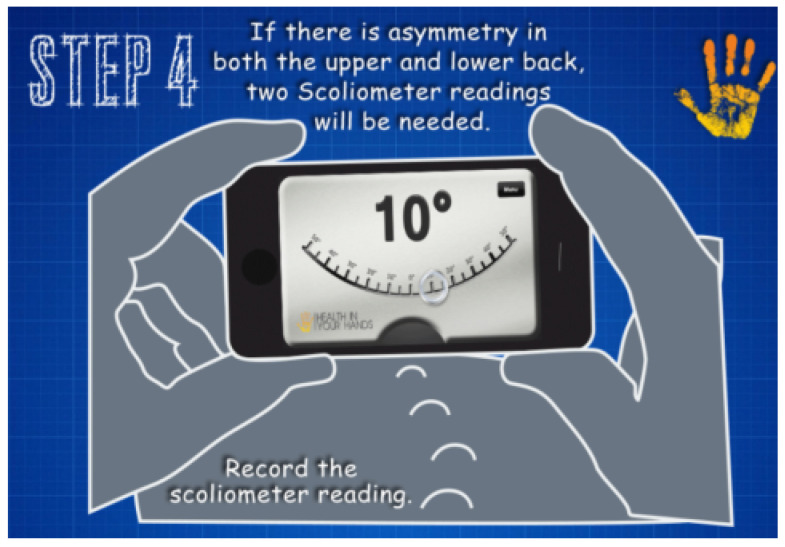
Scoliometer. (This figure is a snapshot taken by the screen of the app).

**Figure 5 ijerph-20-05520-f005:**
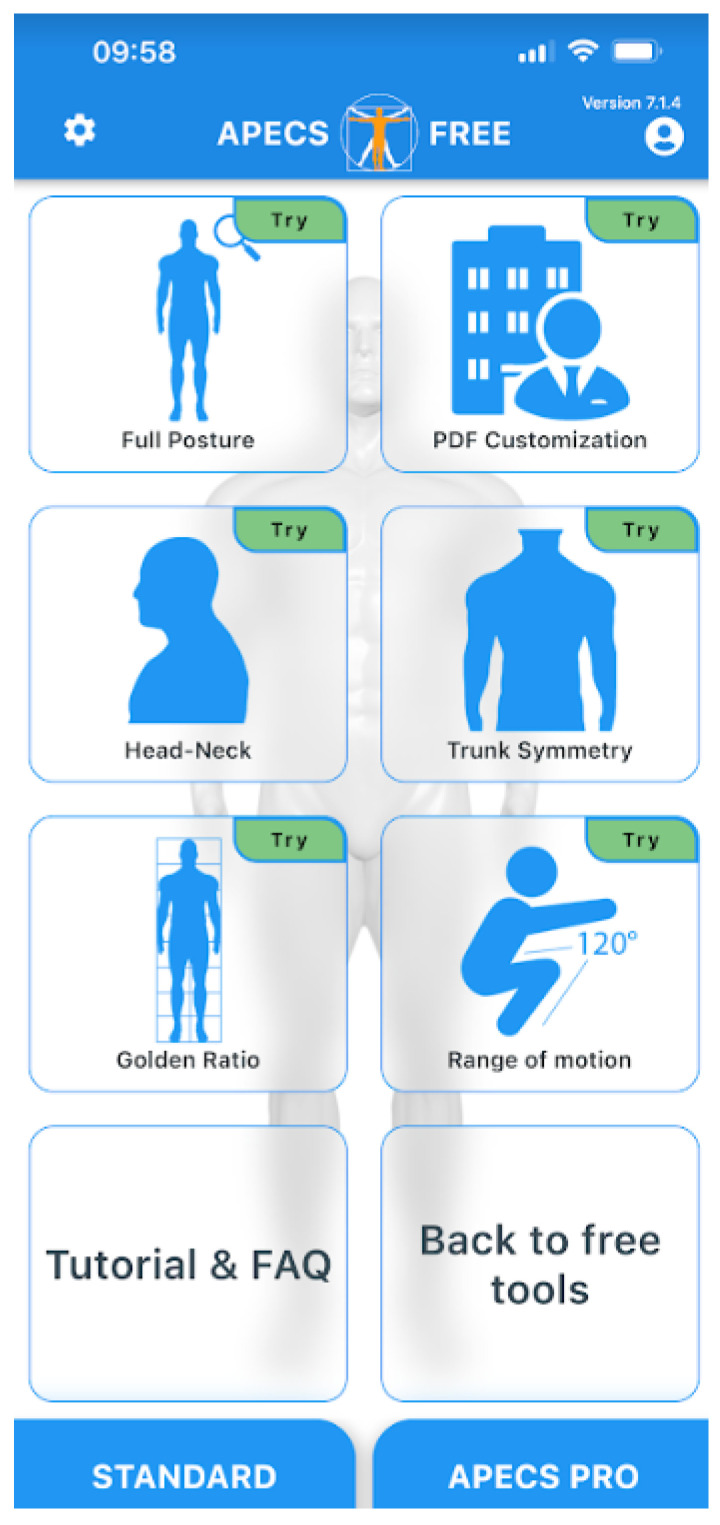
APECS. (This figure is a snapshot taken by the screen of the app).

**Figure 6 ijerph-20-05520-f006:**
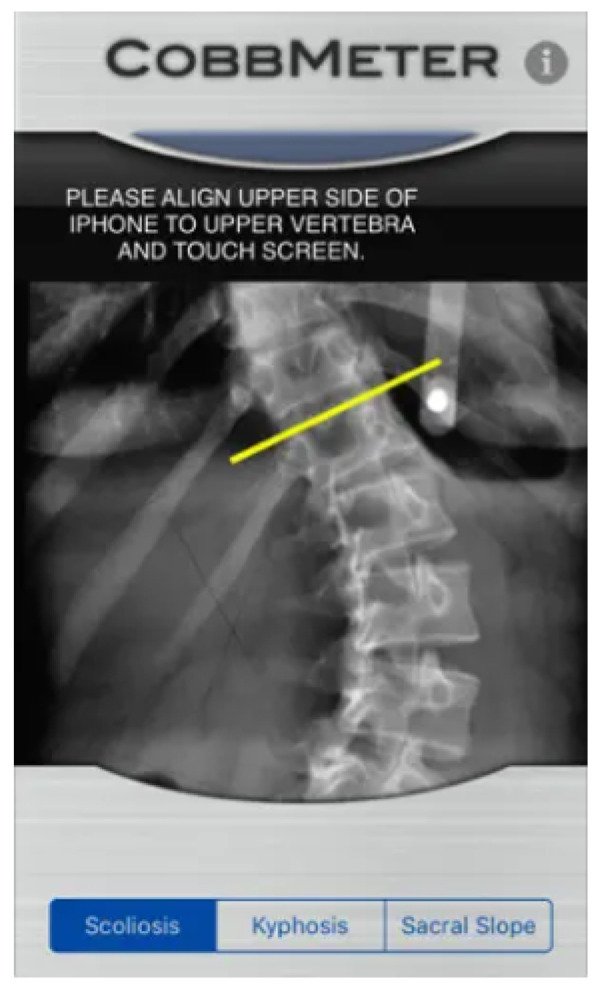
CobbMeter. (This figure is a snapshot taken by the screen of the app).

**Figure 7 ijerph-20-05520-f007:**
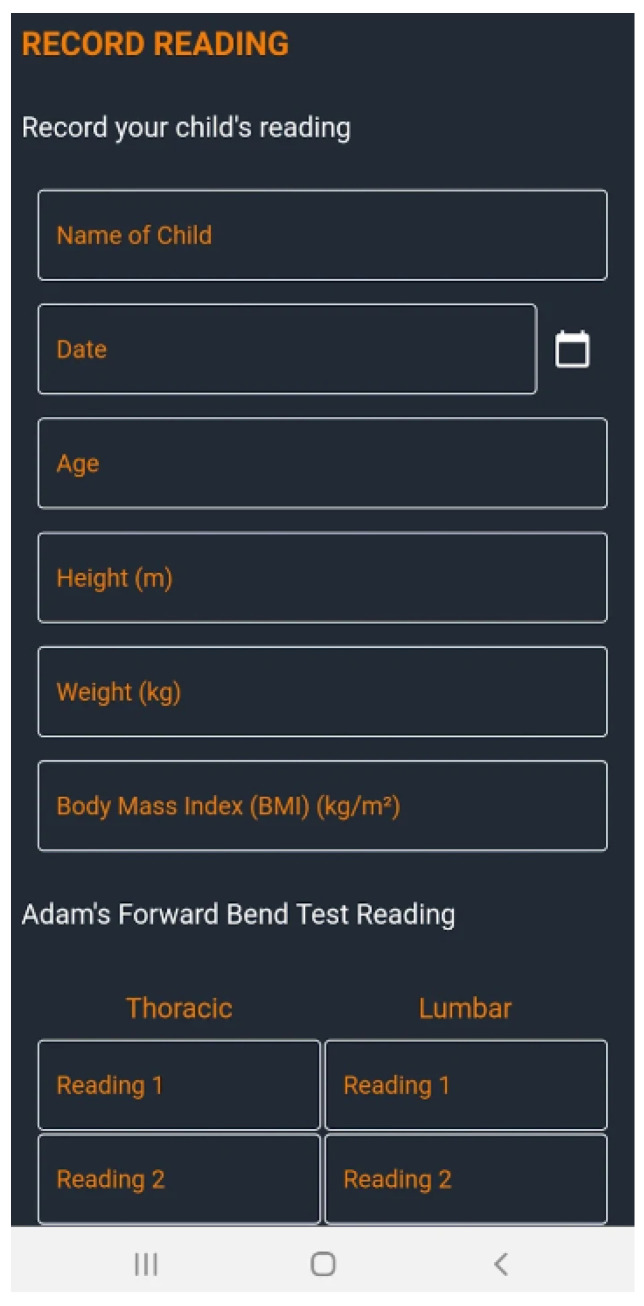
ScolioDetector. (This figure is a snapshot taken by the screen of the app).

**Figure 8 ijerph-20-05520-f008:**
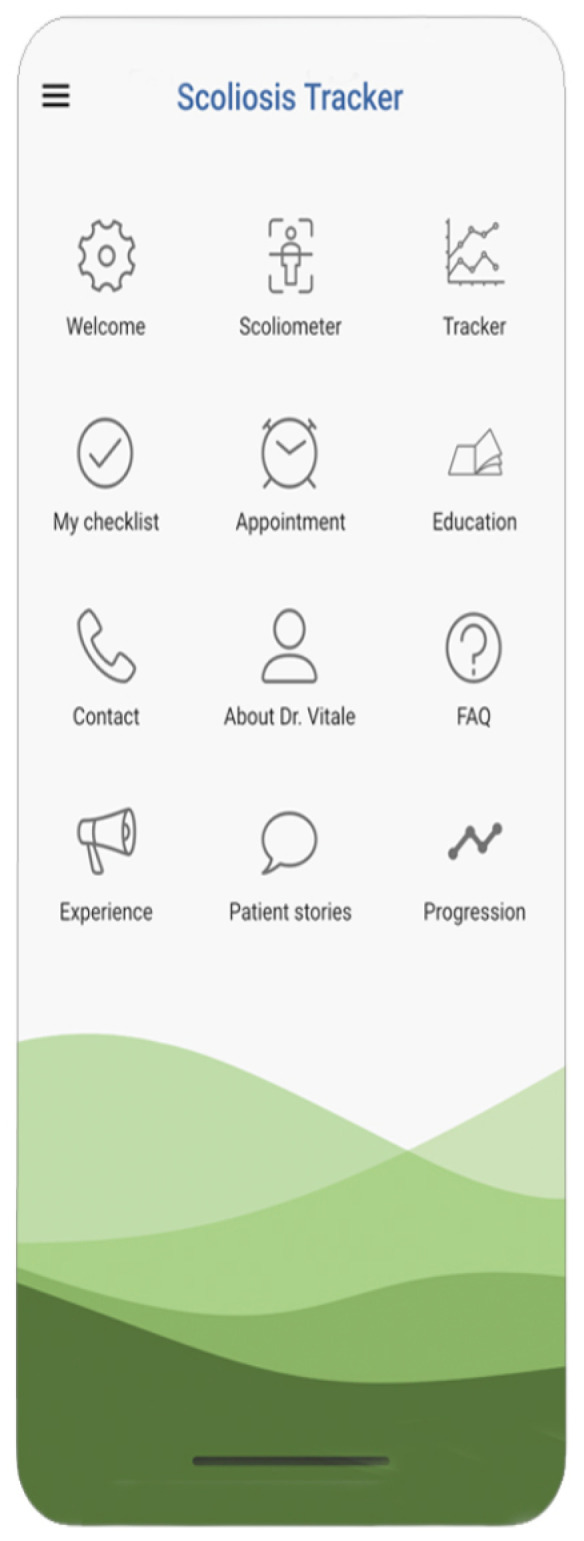
Scoliosis Tracker. (This figure is a snapshot taken by the screen of the app).

**Figure 9 ijerph-20-05520-f009:**
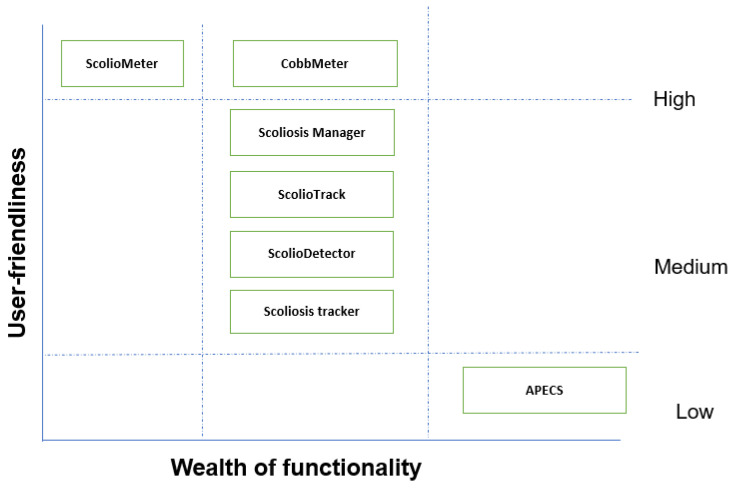
Tool assessment with respect to “Wealth of functionality” and “User-friendliness” dimensions.

**Table 1 ijerph-20-05520-t001:** Comparison of the tools with respect to availability and technology macro-categories. P means pay, F means free, and “-” means not available.

	Availability			Technology	
Name	Reachability	Accessibility	User Support	Operating System	Additional Devices
ScolioTrack	Apple Store, Play Store	P	video tutorial	iOS/Android	None
Scoliometer	Apple Store, Play Store	P	guide	iOS/Android	None
APECS	Apple Store, Play Store	F	video tutorial	iOS/Android	None
CobbMeter	Apple Store	F	-	iOS	None
ScolioDetector	Play Store	F	-	Android	None
Scoliosis Tracker	Apple Store	F	-	iOS	None
Scoliosis Manager	Internet	F	-	Microsoft Windows, Mac OS, Linux	None

**Table 5 ijerph-20-05520-t005:** “Wealth of functionality” rating scale. N indicates the number of functions provided by the tool described in table.

Low	Medium	High
N≤2	3≤N≤5	N≥6

## Data Availability

Data sharing not applicable.

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
