# Peer review of "Scoliosis Management through Apps and Software Tools†"

_ijerph, 2023, doi:10.3390/ijerph20085520_

Round 1

Reviewer 1 Report

Some English formulations deserve re-working to facilitate readers' understanding of the manuscript. The language used leaves a very basic approach, sometimes not being very scientific. The manuscript requires cross-reading by an English native speaker as well as a full grammar / spelling check. Sometimes formulations are very broad and do not provide enough detail.

Just some examples taken from the manuscript:

in the objectives: "best suited to specific needs." <== which needs? Please specify.

 "In fact there is a strong likelihood of curve progression in individuals who present a larger curve and they are young people growing up." <== this sentence is incomprehensible in its logic.

"the first approach is based on the assessment of the spinal curvature thought the radiological imaging" <== thought or throughout?

"There is an incidence of cancer among patients with scoliosis exposed to repeated radiological imaging" <== is it elevated compared to other patients? Is there a relevant correlation?

although it is still user-dependent and much depends on the experience of the observer <== same statement twice

Please rework the language and do re-submit.

Author Response

  1. Some English formulations deserve re-working to facilitate readers' understanding of the manuscript. The language used leaves a very basic approach, sometimes not being very scientific. The manuscript requires cross-reading by an English native speaker as well as a full grammar / spelling check. Sometimes formulations are very broad and do not provide enough detail.

AUTHORS: We thank the Reviewer for this comment.  We asked an English native reader to read and correct the manuscript and we corrected several typos.

Just some examples taken from the manuscript:

  1. in the objectives: "best suited to specific needs." <== which needs? Please specify.

AUTHORS: We thank the Reviewer for this comment. We added the following sentences to underline possible benefits of doctors and patients in using ICT-based solutions:

Benefits for the patients may be: reducing the number of visits to the doctor, self-monitoring of scoliosis. Benefits for the doctors may be: monitoring the scoliosis progression over time, managing several patients in a remote way, mining the data of several patients for evaluating different therapeutic or exercises prescriptions.

  1. "In fact there is a strong likelihood of curve progression in individuals who present a larger curve and they are young people growing up." <== this sentence is incomprehensible in its logic.

AUTHORS: We thank the Reviewer for this comment. We substituted that sentence with the following:

“Indeed, an accurate evaluation of the patient’s growth rate and the risk of curve progression is crucial to undertake the right interventions in a timely manner and only for patients who really need it \cite{choudhry2016adolescent} ”.

  1. "the first approach is based on the assessment of the spinal curvature thought the radiological imaging" <== thought or throughout?

AUTHORS: We thank the Reviewer for pointing this error: The corrected term is “through” and we corrected the sentence.

 "the first approach is based on the assessment of the spinal curvature through the radiological imaging" 

  1. "There is an incidence of cancer among patients with scoliosis exposed to repeated radiological imaging" <== is it elevated compared to other patients? Is there a relevant correlation?

AUTHORS: We thank the Reviewer for this comment. We rephrased the first part of the paragraph with the following paragraph, where we report some references that underline the risks connected with repeated X-ray exposures in scoliosis patients and especially when they are young.

"The radiological examination is the conventional modality to investigate scoliosis. However, the X-ray imaging is strongly related to cancer occurrence due to several radiographic examinations to which the patients are subjected during childhood and adolescence \cite{Nemoto2020ReducingTB}. Because children are more radiosensitive than adults, tools based on tablet PC or smartphone can limit the radiation exposure and thus limit the incidence of such cancers \cite{ahmad2013spinal}} \cite{oakley2019scoliosis}.”

  1. although it is still user-dependent and much depends on the experience of the observer <== same statement twice

AUTHORS: We thank the Reviewer for this comment. We rephrased the sentence as follows:

This procedure has been proven reliable and has less variation than the manual procedure, although it depends on the experience of the observer.

Please rework the language and do re-submit.

Reviewer 2 Report

Many thanks for the opportunity to review this paper entitled “Scoliosis Management through apps and software tools”.

Abstract

Commonly, the management of patients with scoliosis is done through traditional methods or through Information and Communications Technology (ICT) solutions (i.e. softwarebased approaches), that include Smartphone Applications (Apps) and web-based applications.

-        This statement sounds a bit odd. To the best of my limited knowledge of scoliosis most assessments etc are completed in person by a medical professional and the use of ICT is a more recent phenomenon. I think this sentence could be rewritten to improve the clarity of the message.

It is not clear in the abstract – what kind of qualitative approach was used to operationalise this study?

What is the motivation for this research? This is briefly alluded to in Section 3 paragraph 1, however this is very high level.

I know that delays caused by the pandemic and increasing pressures on the healthcare service has increased the patient waiting time to access scoliosis services. Could this be leveraged as an additional motivation to better utilise ICT to support the assessment and management of patients with this diagnosis?

Minor comment

On the other hand,

-        This expression should only be used when preceded by the expression “On one hand”

Typo

-        alrready

-        new methodology p24

Section 3.2

The authors state-

“Many ICT-based methods are too complex, and may be inappropriate for routine clinical evaluation due to their high cost, the difficulty of universal access to the technology, and their immobility (inability to be available everywhere), making their use difficult outside hospitals and medical centers”.

-        Where is the evidence to support this statement?

Section 3

How are the technologies presented in this article selected? Is this an exhaustive collection of web-based and app-based technologies? Please provide a more detailed rationale for the inclusion of these.

Section 4

This section presents a set of 5 macro categories as a qualitative evaluation approach for this study.

First, it would be helpful if the authors could provide a rationale for the evaluation of the technologies presented, why is this an important thing to do? From a clinician, patient perspective?

Second, where do the 5 categories presented come from? Existing literature? The authors?

Did the authors consider other way to evaluate these technologies? For example, Brookes (1996) System Usability Scale.

Brooke, J. (1996). SUS-A quick and dirty usability scale. Usability evaluation in industry189(194), 4-7

Conclusion

The conclusions present the advantages of the both the methodology proposed and the use of technology in the assessment and management of scoliosis from both a doctor and patient perspective. Have the team considered the limitations of the evaluation approach; the challenges associated with the ICT presented?

What are the limitations of this work? How does it contribute to existing research and practice? What are the opportunities for future research in this area?

The use of technology in the area of scoliosis is an important and under researcher one. This study has merit, however in its current format it has many limitations which I have pointed to above. I suggest that a major revision is required before this article may be considered for publication.

Author Response

Many thanks for the opportunity to review this paper entitled “Scoliosis Management through apps and software tools”. 

Abstract

Commonly, the management of patients with scoliosis is done through traditional methods or through Information and Communications Technology (ICT) solutions (i.e. softwarebased approaches), that include Smartphone Applications (Apps) and web-based applications.

  1. This statement sounds a bit odd. To the best of my limited knowledge of scoliosis most assessments etc are completed in person by a medical professional and the use of ICT is a more recent phenomenon. I think this sentence could be rewritten to improve the clarity of the message.

AUTHORS: We thank the Reviewer for this comment. We rephrased the sentence as follows:

“Commonly, the scoliosis evaluation is done by the medical professionals in person using traditional methods (i.e. scoliometer and/or X-ray radiographs). In recent years, as happened in various medicine disciplines, also in orthopedics it is possible to observe the spread of the Information and Communications Technology (ICT) solutions (i.e. software based approaches). As an example, Smartphone Applications (Apps) and web-based applications may help the doctors in screening and monitoring scoliosis reducing the number of in person visits.”

  1. It is not clear in the abstract – what kind of qualitative approach was used to operationalise this study?

AUTHORS: We thank the Reviewer for this comment. We rephrased the sentence as follows:

“We first propose a methodology for the scoliosis apps evaluation that considers fives macro-categories: i) technological aspects (e.g. available sensors, how angles are measured); ii) the type measurements (e.g. Cobb angle, Angle of Trunk Rotation, Axial vertebral rotation); iii) availability (e.g. app store and eventual fee to pay); iv) the functions offered to the user (e.g. posture monitoring, exercise prescription); v) overall evaluation (e.g. pros and cons, usability).”

  1. What is the motivation for this research? This is briefly alluded to in Section 3 paragraph 1, however this is very high level. I know that delays caused by the pandemic and increasing pressures on the healthcare service has increased the patient waiting time to access scoliosis services. Could this be leveraged as an additional motivation to better utilise ICT to support the assessment and management of patients with this diagnosis?

AUTHORS: We thank the Reviewer for this comment and the suggestions. We explicitly stated the motivation in the Introduction, adding the following paragraph.

The COVID-19 pandemic had a heavy impact on the public healthcare systems. In particular, the access to health services has been compromised and, as a result, delays in the scoliosis diagnosis occur frequently \cite{COVID_19}. Delayed diagnosis of scoliosis can increase the patient’s risk of requiring surgical intervention. The spread of scoliosis screening systems that uses Apps and web-based applications can facilitate the early detection of scoliosis on a global scale mitigating the adverse impact of the COVID-19 pandemic on the diagnosis, management of the scoliosis.

The increased availability of such apps and the great pressure to use ICT-based solutions at home due to COVID-19, is a main reason to provide guidelines for the adoption of apps by the doctors and patients. Thus, the main motivation for this work is to provide a guideline in the choice of scoliosis management apps, that considers both doctors needs (e.g. which angle the app measures, if the doctor can prescribe physical exercises at home, etc.), and patients needs (e.g. easiness of use, availability on app store, eventual fee to pay, etc.).

  1. Minor comment

On the other hand,

-        This expression should only be used when preceded by the expression “On one hand”

AUTHORS: We thank the Reviewer for this comment. We rephrased the sentence as follows.

“The traditional scoliosis management comprises two main activities: i) the assessment of the  spinal curvature \textcolor{blue}{through} the radiological imaging; and ii) the physical examination of the patient which employs a goniometer (scoliometer) to follow and estimate the angular deformity of the spine.  “

Typo

-        alrready

-        new methodology p24

 AUTHORS: We thank the Reviewer for this comment. We fixed the typos.

  1. Section 3.2 - The authors state-“Many ICT-based methods are too complex, and may be inappropriate for routine clinical evaluation due to their high cost, the difficulty of universal access to the technology, and their immobility (inability to be available everywhere), making their use difficult outside hospitals and medical centers”.   Where is the evidence to support this statement?

AUTHORS: We thank the Reviewer for this comment. We rephrased the first part of section 3.2 as follows.

Due to the spread of the COVID-19 pandemic, ICT has been utilized to a much larger extent to limit the contact between persons. Therefore, a greater number of people became familiar with the ICT technology reporting high levels of satisfaction \cite{pmid33089712}. In this context, the smartphone has become the most widely used ICT-based system in the world and it has been gradually introduced in the clinical practice. 

Indeed, smartphones are capable to provide healthcare services anytime and anywhere in a cheap way. Indeed, smartphones can replicate the function of traditional medical devices such as the scoliometer, saving cost and time (e.g. for stable cases to lengthen the period between follow-ups), thus improving clinical efficiency and convenience \cite{pmid23147635}.

  1. Section 3 - How are the technologies presented in this article selected? Is this an exhaustive collection of web-based and app-based technologies? Please provide a more detailed rationale for the inclusion of these.

AUTHORS: We thank the Reviewer for this comment. We substituted the sentence 

“In the following, main APPs for scoliosis measurement and management are presented.”

with the following paragraph:

In recent years, a plethora of ICT-based solutions for healthcare have been developed and there is an ongoing development of new ICT tools and methods, including Apps and web-based applications. To select the apps to evaluate, we performed an extensive search by analysìzing both the literature discussing scoliosis management through apps \cite{mobile_pps} and by consulting the web. Moreover, some apps were suggested by the orthopedist participating in this work. Some apps that were cited in the literature but unavailable on the store were excluded. Thus, we are confident that the systems discussed in the following are the most popular apps for scoliosis management.

  1. Section 4 -This section presents a set of 5 macro categories as a qualitative evaluation approach for this study.  First, it would be helpful if the authors could provide a rationale for the evaluation of the technologies presented, why is this an important thing to do? From a clinician, patient perspective?

AUTHORS: We thank the Reviewer for this comment. We added the following paragraph at the end of Section 4.

The main rationale for the definition of an evaluation methodology is to have a methodological framework that can be used not only in this work to evaluate the current apps, but also in future works when new apps are available. As described previously, the proposed methodology for the scoliosis apps evaluation considers fives macro-categories that are relevant either to the patients (e.g. the availability or the cost, the possibility to receive prescriptions of exercises at home), or to the clinician (e.g. some technical features such as the angles that the app can measure or the possibility to monitor remotely the patients or to prescribe them exercises at home).

Moreover, considering that several apps for scoliosis management have been developed and publicly available, choosing which app is the right fit for the user can be a time-consuming task and it can lead to confusion, we also provide a simple guideline to choose the apps.Having the previous methodology helps in compiling the guideline.

  1. Second, where do the 5 categories presented come from? Existing literature? The authors? Did the authors consider other way to evaluate these technologies? For example, Brookes (1996) System Usability Scale.Brooke, J. (1996). SUS-A quick and dirty usability scale. Usability evaluation in industry189(194), 4-7

AUTHORS: We thank the Reviewer for this comment. We added the following paragraph at the end of Section 4.

As described previously, the 5 macro-categories consider: i) technological aspects (e.g. available sensors, how angles are measured); ii) the type measurements (e.g. Cobb angle, Angle of Trunk Rotation, Axial vertebral rotation); iii) availability (e.g. app store and eventual fee to pay); iv) the functions offered to the user (e.g. posture monitoring, exercise prescription); v) an overall evaluation (e.g. pros and cons, usability).

Such categories come from our experience in the analysis of other ICT-based tools and apps in the bioinformatics \cite{BMC:PPI:Visualization} and biomedical fields \cite{https://doi.org/10.1002/widm.1090}.

Although those surveys have investigated tools in different domains with respect to the scoliosis management, those tools share with scoliosis management apps the fact that the user is not an ICT expert (e.g. clinicians or bioinformaticians) and the fact that they manages biomedical and clinical data.   

Moreover, we are aware of the System Usability Scale (SUS), that is a methodology for measuring the usability (i.e the subjective perception of interaction between the user and the system) of a wide variety of products and services, including hardware, software, mobile devices, websites, and applications. 

Although the SUS has many benefits that make it popular for the usability assessment, it has many limits. For instance, SUS primarily measures the perceived usability of a system rather than the system overall usability \cite{10.1007/978-3-319-91797-9_25, doi:10.1177/1541931213571043}. Moreover, although SUS covers several aspects related to the usability, such as complexity, learnability, however it is unable to provide accurate information on the system weakness. 

Moreover, recent studies about SUS have reported that users having a more extensive experience with a system tended to provide higher and more favorable evaluation with respect to users with no or limited experience with the system (i.e. it can be affected by bias). 

Furthermore, SUS omits important aspects that are specific for the type of system (e.g. fashion and ergonomic design for the wearable healthcare tools).

Our proposed methodology based on the 5 categories goes beyond the SUS usability measurement to provide an overall assessment of the ICT scoliosis tools. Indeed, it covers technological aspects (e.g. Operating System, Additional Devices), measurement (e.g. type of measurement), qualitative evaluation (e.g. strengths and weaknesses) that are not included in the SUS evaluation.

Conclusion - The conclusions present the advantages of the both the methodology proposed and the use of technology in the assessment and management of scoliosis from both a doctor and patient perspective. Have the team considered the limitations of the evaluation approach; the challenges associated with the ICT presented? 

AUTHORS: We thank the Reviewer for this comment. We added the following paragraph in Conclusions.

A limitation of the proposed methodology is related to the number of assessed characteristics: although we tried to cover many characteristics of scoliosis management apps, it is possible that some important characteristics have been overlooked. As future work we intend to overcome this limitation by extending the number of aspects and/or categories 

  1. What are the limitations of this work? How does it contribute to existing research and practice? What are the opportunities for future research in this area?

AUTHORS: We thank the Reviewer for this comment. We added the following paragraphs in Conclusions.

The main limitation of this work is related to the apparently low number of the assessed tools in the study. As future work we intend to overcome this limitation by extending the number of the ICT-based scoliosis tools that will be compared. Furthermore, we plan to develop an app that brings additional functionalities with respect to current apps, on the basis of the requirements provided by the orthopedic participating in this study.

Finally, the contribution of this paper to existing research and practice is manifold: first, it proposes a methodology for the systematic assessment of ICT-based scoliosis tools; second, it provides the researchers with an evaluation of the main technical and functional features of the most popular apps and web-based applications for scoliosis; third, it provides an easy to use guideline that aims to help the doctors, specialists and those who wish to perform scoliosis checks at home, in the choice of the ICT-based scoliosis tools, underling several aspects of interests for medical professionals and patients.

In conclusion, the use of apps in scoliosis management, that allows the extensive collections of clinical data, besides easing clinical practice, opens the doors to the future application of machine learning and artificial intelligence in orthopedics.

The use of technology in the area of scoliosis is an important and under researcher one. This study has merit, however in its current format it has many limitations which I have pointed to above. I suggest that a major revision is required before this article may be considered for publication.